# Application of Permeation Enhancers in Oral Delivery of Macromolecules: An Update

**DOI:** 10.3390/pharmaceutics11010041

**Published:** 2019-01-19

**Authors:** Sam Maher, David J. Brayden, Luca Casettari, Lisbeth Illum

**Affiliations:** 1School of Pharmacy, Royal College of Surgeons in Ireland, St. Stephens Green, Dublin 2, Ireland; 2School of Veterinary Medicine and UCD Conway Institute, University College Dublin, Belfield, Dublin 4, Ireland; david.brayden@ucd.ie; 3Department of Biomolecular Sciences, University of Urbino Carlo Bo, Piazza del Rinascimento 6, 61029 Urbino (PU), Italy; luca.casettari@uniurb.it; 4IDentity, 19 Cavendish Crescent North, The Park, Nottingham NG7 1BA, UK; lisbeth.illum@illumdavis.com

**Keywords:** permeation enhancer, oral delivery, formulation, permeability, safety, simulated intestinal fluid, hydrophobization, epithelium

## Abstract

The application of permeation enhancers (PEs) to improve transport of poorly absorbed active pharmaceutical ingredients across the intestinal epithelium is a widely tested approach. Several hundred compounds have been shown to alter the epithelial barrier, and although the research emphasis has broadened to encompass a role for nanoparticle approaches, PEs represent a key constituent of conventional oral formulations that have progressed to clinical testing. In this review, we highlight promising PEs in early development, summarize the current state of the art, and highlight challenges to the translation of PE-based delivery systems into safe and effective oral dosage forms for patients.

## 1. Introduction

Since the early 1990s, the pharmaceutical industry has gradually reduced attrition related to non-optimal pharmacokinetics (PK) and low bioavailability (BA) of drugs in development [1]. A combination of tools that predict sub-optimal physicochemical properties, as well as technologies that address impediments to translation from in vitro/preclinical studies to successful clinical trials (e.g., low aqueous solubility, short plasma half-life (t½)) has led to a shift in overall attrition from PK to safety. Despite this, non-optimal PK characteristics continue to rank amongst the most common cause of attrition [2]. The shift towards the development of more lipophilic compounds that are less likely to suffer from sub-optimal PK has however been associated with low solubility and increased toxicity [3,4]. A cohort of macromolecules that can exhibit desirable safety and efficacy are large hydrophilic compounds with greater molecular complexity (e.g., oligomeric peptide backbones), but such properties inevitably confer low intestinal epithelial permeability. In theory, poorly permeable drugs may be administered orally at higher doses to offset low absorption, but this is usually impractical for formulation and cost reasons. Low permeability might be acceptable if a safe and consistent therapeutic effect can be achieved with a molecule that has a long t½, even in the context of high variability in BA, otherwise poorly permeable drugs are typically formulated in injectable dosage forms or in dosage forms using other routes of delivery with higher membrane permeability than the intestine, such as the nasal route.

A paucity of delivery technologies that address low intestinal epithelial permeability for macromolecules has left pharmaceutical manufacturers with little option but to limit screening of these complex hydrophilic macromolecules or default to parenteral formulation. Delivery systems that enable poorly permeable molecules to be efficiently delivered across the intestine may diversity the type of compounds screened in discovery and may permit reformulation of selected injectable macromolecules. There is debate as to whether medicinal chemists in the discovery field should solely focus on safety and efficacy of the active, and rely on formulation and delivery scientists to address sub-optimal solubility, ADME (absorption, distribution, metabolism and excretion), and stability characteristics—or whether medicinal chemists should focus on all of the aforementioned properties to rely less on delivery and formulation scientists [5]. Recent advances highlight the importance of dual focus early in development: optimizing the molecule and the formulation [6,7]. The key challenge with focusing on delivery platforms is the current lack of proven technologies that can significantly improve intestinal permeability in humans. 

## 2. Permeation Enhancer (PE) Categories

The potential of improving the oral BA of macromolecules has been extensively researched, but the majority of delivery systems have failed to progress beyond preliminary animal model evaluation. The inclusion of an excipient that facilitates transport across the intestinal epithelial barrier is a desirable approach. Proprietary formulations that attempt to improve oral absorption of macromolecules in humans usually include permeation enhancers (PEs). However, failure of these PE-based formulations to translate into commercial products has led academic investigators to prioritize development of more technologically advanced drug delivery systems rather than address impediments to translation of PEs. Several hundred compounds have been shown to alter permeability in oral, nasal, buccal, pulmonary, vaginal, and corneal delivery models. These compounds are broadly categorized as paracellular or transcellular PEs.

Paracellular PEs can be sub-categorized into first and second generation [8]. The older first generation paracellular PEs open tight junctions (TJs) through intracellular signaling mechanisms, while the second sub-category act via direct disruption of homophilic interactions at cell adhesion recognition (CAR) sequences. Major targets to afford TJ opening include: the cytoskeleton, claudins, occludin, and E-cadherin and Ca^2+^ [9]. Toxins and their derivatives are one of the largest sources of paracellular PEs, irrespective of sub-category (e.g., *zonula occludins* toxin, *clostridium perfringens* enterotoxin (CPE) [10]). Despite extensive preclinical assessment of microbial toxins and derived motifs, clinical trials with paracellular PEs are dominated by older agents with broader mechanisms of action, one example being ethylenediaminetetraacetic acid (EDTA) (POD^TM^, Oramed, Israel [11]).

Transcellular PEs alter epithelial permeability by two contrasting mechanisms, (i) reversible perturbation of the epithelial plasma membrane [9], or (ii) physical interaction with the active to improve passive transcellular permeation (e.g., hydrophobization [12]). Surfactant-based PEs are a widely tested category that alter membrane integrity. Included in this category are medium chain fatty acids, acylcarnitines, acylated amino acids, bile salts, and a variety of non-ionic surfactants (e.g., polyoxyethylene-8 lauryl ether (C_12_E_8_), sucrose laurate, macrogol-8 glycerides [13,14]). A number of surfactants have been evaluated clinically in oral delivery systems for macromolecules: lauroylcarnitine chloride (Peptilligence^TM^, Enteris Biopharma, Boonton, NJ, USA [15]) sodium caprate (C_10_) (GIPET^TM^, Merrion Pharma, Dublin, Ireland [16]), sodium caprylate (TPE^TM^ Chiasma, Ness Ziona, Israel [17]), and sodium cholate (Biocon, Bangalore, India [18]). Soluble and insoluble surfactants are also constituents of complex lipoidal systems including oily suspensions [17] and emulsions [19]. At low test concentrations in reductionist cell and tissue based delivery models, transcellular perturbants (i) activate plasma membrane receptors and enzymes, (ii) modulate intracellular mediators, (iii) selectively remove TJ proteins from fluidic regions of the membrane, and (iv) initiate repair mechanisms related to opening of TJs [20]. In some cases, these actions are uncoupled from membrane perturbation [21]. This has led investigators to suggest that some perturbants may in part act indirectly via a paracellular mode of action. However, low concentrations of such agents that do not induce transcellular perturbation cause only modest increases on permeability in vitro [21]. 

Transcellular permeation may also be improved by physical complexation, either by hydrophobic ion pairing (HIP) or dipole–dipole interaction [9]. HIP involves electrostatic-based complexation of an ionizable lead (usually a peptide) with an amphiphilic counterion. The hydrophobic moiety of the counter ion confers a lower capacity for solvation than conventional counterions typically used in the preparation of pharmaceutical salts to address low aqueous solubility. HIP reduces the solubility of several peptides including insulin [22], desmopressin [23], octreotide [24], and exenatide [25]. Hydrophobization via dipole–dipole interactions between the poorly permeable macromolecule and acylated amino acids (the so-called Eligen^®^ carriers of Emisphere, Roseland, NJ, USA [26]) is a more widely studied approach than HIP, although the less well understood. Emisphere have assessed the clinical potential of Eligen carriers most notably SNAC (sodium salcaprozate) and 5-CNAC (N-(5-chlorosalicyloyl)-8-aminocaprylic acid) over a 20-year period. In that time, Emisphere discontinued development of SNAC for oral delivery of heparin and insulin. SNAC has however been successfully used in a marketed oral vitamin B_12_ supplement (Eligen B_12_) [27], and more recently was shown to improve oral absorption of semaglutide in Phase II trials [28]. Development of an oral salmon calcitonin (sCT) using 5-CNAC failed to meet primary endpoints in two Phase III trials [29].

Several non-surfactant PEs have also been tested in pre-clinical studies. These include chitosan and its derivatives, cell penetrating peptides (CPPs), solvents (e.g., ethanol), salicylates, and enamines. CPPs such as penetratin and its analog, PentraMax^TM^, continue to be researched for oral peptide delivery. There is evidence that these CPPs act by altering membrane barrier integrity [30], endocytosis [30], and physical complexation [31]. Although a few CPPs have progressed to clinical evaluation, the majority relate to the intracellular delivery of small molecules and not to oral delivery of macromolecules [32]. It remains to be seen if CPPs will eventually advance to clinical testing in oral delivery of anti-diabetic peptides [33].

## 3. Targets for Intestinal Permeation Enhancement: Beyond Insulin

Development of delivery platforms that improve epithelial permeability was historically associated with creating non-invasive formulations of insulin. Insulin represents an inexpensive and available prototype peptide with established analytical methods for PK and pharmacodynamic assays. In justifying the use of insulin, it can be argued that a prototype that can improve permeation of this large peptide (5.8 kDa) could be even more effective with smaller peptides (1–4 kDa), so it is a high bar. While there is some merit to the development of an oral insulin dosage, the focus on insulin has restricted effort to develop oral delivery systems for other macromolecules with more favourable physicochemical properties. Additionally, the emphasis on oral delivery of insulin and the lack of success in that pursuit during the hype of the 1990s has led to a largely negative view in the pharmaceutical industry and with journal editors of novel strategies to improve intestinal permeability.

Table 1 shows a selection of licensed peptides marketed via oral or injectable routes. This table shows that dose (potency), t½, Mw, lipophilicity (LogP) and target action site are important factors that influence whether a peptide is commercially successful via oral or injectable routes. There are currently nine peptides on the market as oral products [34], although only two desmopressin and cyclosporin are intended to act on systemic targets; the properties of which are shown in Table 1. All other peptides in Table 1, except the regionally acting vancomycin, are marketed in injectable formulations and BA values are considered negligible (<0.1%).

All peptides marketed via the oral route are assigned low permeability within the Biopharmaceutics Classification System (BCS). Marketed injectable peptides are not formally assigned a BCS Class, although all peptides in Table 1 are considered BCS Class III. It is evident from Table 1 that while peptides exhibit low and variable BA (apart from cyclosporin), an optimal balance between potency, size/complexity, and apparent plasma t½ can facilitate the development of a successful oral form (e.g., desmopressin) (Table 1). The extent to which chemical modification and the delivery system can compensate for peptides that do not exhibit an optimal balance between potency, size/complexity, and apparent plasma half-life is not clear. Structural modifications can reduce demand on delivery systems for peptides that are larger and more complex. A recent example being the very long t½ GLP-1 analog, semaglutide [35], which has demonstrated promising results in clinical trials in an oral formulation containing SNAC, a PE that failed with macromolecules that did not have such favorable features.

In atypical cases where peptides are lipophilic and stable to proteolytic degradation, oral BA can be significantly higher than equivalent hydrophilic peptides (e.g., cyclosporin). The downside of high lipophilicity is low aqueous solubility, which has been addressed using lipid-based formulations [36]. Clinical evaluation of the cyclosporin (Sandimmune^®^ Neoral, Novartis) has shown that 86% of the intact peptide permeates the intestinal epithelium [37]. While brush border and hepatic metabolism reduce BA of cyclosporin to 27%, oral BA is far higher than equivalent hydrophilic peptides such as desmopressin and octreotide. Chemical modification has been less successful in addressing low permeability, for example, in the case of insulin [38], although there is emerging evidence that physical modification of peptides using hydrophobizing agents may improve passive permeation (Section 4.7) [9,12,39]. A key area of development, therefore, is the combined development of hydrophobized peptides with a suitable lipid-based delivery system.

## 4. Recent Highlights

### 4.1. Oral Semaglutide Reduces HBA1c in Type 2 Diabetics by over 1.5% in Phase II Trials

Novo Nordisk (Bagsværd, Denmark) has adopted an integrated approach to enabling oral peptide delivery, combining structural engineering and formulation optimization [6]. Semaglutide is a long-acting acylated GLP-1 agonist (Ozempic^®^ once weekly, s.c.) that also has greater stability to enzymatic degradation in the gastrointestinal (GI) tract. A once-a-day oral formulation of semaglutide in an immediate release formulation with SNAC is likely to have a significant effect on the management of diabetes. In a Phase II randomised controlled trial of 632 patients, daily administration of oral semaglutide (2.5–40 mg) lowered HbA1c by 0.7% to 1.9% [28]. Although there was a 280-fold difference in the cumulative dose relative to the once weekly sub-cutaneous (sc) injection (1 mg), the long plasma t½, high potency, and improved stability reduces the need for a high-performing PE. Induction of high local pH in the immediate vicinity of the semaglutide/SNAC tablet in the stomach to increase solubility is considered a central aspect of the technology [41], where the contemporaneous release of SNAC and semaglutide to fasting patients enables co-localisation in high concentration at the site of enhancement. In gamma scintigraphy studies in healthy volunteers, the anatomical location for tablet erosion was confirmed as the stomach irrespective of whether participants ingested the formulation with 50 mL or 240 mL water [42]. The time to complete tablet erosion was 95 min (50 mL water) and 66.2 min (240 mL). Slow erosion of tablets (<54% in 1 h) was associated with higher plasma semaglutide levels and a longer T_max_ compared to fast eroding tablets. It is unclear why gastric emptying does not occur in patients receiving oral semaglutide, although it is possible that the peptide could slow gastric emptying [43]. It is not clear if the slow progressive release of semaglutide in the stomach may necessitate a longer delay before ingestion of a meal. The unusual gastric pH mechanism advocated for semaglutide/SNAC seems to be highly specific for this peptide. This topic is discussed in detail in the current special issue review by Twarog et al.

### 4.2. The Ionic Liquid Choline Geranate (CAGE) Has a Major Effect on Oral BA of Insulin

Ionic liquids are salt-like materials that are liquids below 100 °C. There has been renewed interest in the use of ionic liquids primarily due to their solvent properties. A recent study showed that intestinal co-instillation of a relatively low dose of insulin with choline geranate (CAGE) led to a dramatic lowering in blood sugar levels in non-diabetic rats [44]. The decrease in blood sugar compared favourably with the s.c. delivered insulin. The oral BA of insulin (10 IU) in rats was 45% following delivery with CAGE (80 mg) in enteric coated capsules in rodents, one of the highest values ever recorded for an insulin formulation in a rat PK study, albeit relative to 2 IU (s.c.) as against 1IU (s.c.) in most studies. CAGE increased the fluidity of mucous, reduced epithelial barrier integrity and protected insulin from degradation by trypsin. There was no apparent histological damage to the intestinal mucosa of rats. However, CAGE caused a partial reduction in Caco-2 cell viability, a decrease in transepithelial electrical resistance (TEER) and evidence of intracellular insulin-fluorescein isothiocyanate (FITC) in fluorescence microscopy, suggesting some transcellular perturbation. It is noteworthy that choline-based ionic liquids were amongst the least cytotoxic in a screen assessing the effect of a panel on the viability of Caco-2 and HT29 cells [45]. Moreover, insulin was stable in CAGE for four months at 4 °C. It is noteworthy that peptide stability concerns have been overcome in non-aqueous polyprotic solvents [46]. It will be interesting to see if these results translate to large animal models with a dosage form that can translate to human trials.

### 4.3. Mode of Action Studies on the PE, PIP 640

Permeant inhibitor of phosphatase (PIP) peptide 640 (PIP 640) is a decapeptide that transiently opens TJs. PIP640 inhibits myosin light chain phosphatase (MLCP), which in turn inhibits dephosphorylation of myosin light chain (MLC) [47]. This is achieved by binding to a subunit complex of protein phosphatase 1 (PP1) and MYPT1 (myosin phosphatase target subunit) in the same manner as CPI-17 (C-kinase-activated protein phosphatase-1 (PP1) inhibitor-17kDa). PIP-640 increased fluorescein isothiocyanate dextran 4 kDa (FD4) permeation across Caco-2 monolayers and improved BA of insulin to 4% in rat intestinal instillations [47]. More recently, a structure–activity assessment of PIP 640 showed that residues of glutamic acid and tyrosine are requisite for binding to MYPT1 subunit of the MLCP complex, while substitution of aspartic acid for arginine led to more specific targeting of PP1 subunit, which was associated with greater cytotoxicity [48]. In a follow-on study, PIP 640 was shown to selectively increase total levels of claudin 2 (both the cytoplasm and at the membrane) [49]. This increment was attributed to the preserved stability of existing claudin 2 rather than increased expression. Alteration of Caco-2 monolayer permeability by PIP 640 was biased towards paracellular transport of positively-charged diethylaminoethyl dextran compared to neutral dextran or carboxymethyl dextran, in keeping with data showing that claudin-2 is responsible for the formation of a channel that is selective for cations. A similar effect was observed in the comparison of sCT (cationic at the pH in the small intestine) and exenatide (anionic at the pH in the small intestine). PIP640 caused a greater increase on permeation of sCT over exenatide in both Caco-2 monolayers and rat intestinal instillations. Together these data highlight a potential for development of new chemical entity PEs that show bias towards permeation of cationic actives. These studies are noteworthy due the molecular biology and rational screening approaches taken to produce a PE molecule. Increasing the potency of PIP 640 is however likely to be required in order to allow a translatable oral dosage formulation to be developed.

### 4.4. Application of Nanoparticles to Co-Localise Active and PE

Nanotechnology has several potential applications in the science of oral delivery including protecting payload, targeting epithelial receptors in GI regions, and improving permeability [50]. The original working hypothesis for application in oral delivery of peptides was that nanoencapsulation would protect labile actives from pre-systemic degradation and shuttle cargo across the intestinal epithelium [51]. While this is the desired outcome, low and variable transmucosal uptake of nanoparticles and formulation complexity continue to impede progression. Investigators have attempted to improve transmucosal flux of drug-loaded nanoparticles using PEs, but it is difficult to envisage a PE improving uptake of colloids when uptake of peptides is inherently low and variable. An evolving view of nanoencapsulation is their potential to permeate mucus and to co-localise release of the macromolecule and PE at the intestinal epithelial wall, a key requirement for effective permeation enhancement [52,53]. It remains to be seen whether PE-macromolecule loaded nanoparticles, either passive or receptor-targeted, can be an effective strategy for delivering payloads across the intestine. 

### 4.5. Application of PEs in Delivery of Nutraceuticals

Bioactive molecules in foodstuffs may have potential health benefits beyond their basic nutritional value. Many of these are complex natural substances that have low aqueous solubility and/or low intestinal permeability. These include peptides, carbohydrates, lipids, and complex organic phytochemicals. There is considerable overlap in the approaches to enable oral delivery of pharmaceutical and nutraceutical products [54], although development considerations related to safety, efficacy, and cost-effectiveness vary between the two.

Application of PEs to improve the oral BA of nutraceuticals is an emerging area. Safety concerns related to additives that might alter intestinal barrier integrity may outweigh any health benefits derived from selected nutraceuticals. PEs that are likely to be useful in oral delivery of nutraceuticals are substances that have food additive status, GRAS status, or can be made to food grade, so this limits the range for selection. These include medium and long chain fatty acids, bile salts (e.g., sodium chloeate [13]), chitosan and its derivatives [55] and certain non-ionic surfactants (e.g., sucrose esters [56], lactose esters [57], polysorbates [58]). Although most PEs are tested with transport markers of poorly permeable drugs, recent studies have evaluated oral delivery of antihypertensive tripeptides derived from milk and chicken muscle. Co-delivery of C_10_ (180 mM) with isoleucine-proline-proline (IPP) and leucine-lysine-proline (LKP) from milk did not improve oral absorption any further in rats in part because basal permeability seemed to be already high [59]. C_10_ reversed the effect of the Pep-T1 inhibitor (glycyl sarcosine) on oral absorption of these tripeptides. Other PEs have been shown to act in part via solubilization (bile salts [60]), inhibition of transporters (piperine [61]), or alteration to pre-systemic metabolism (genistein [62]). There are questions as to whether PEs can be effective as part of food matrices, and it may, therefore, be necessary to deliver bioactives as nutritional supplements in capsules or tablets. The relatively lower potency of nutraceutical versus therapeutic peptides is a challenge to the use of PEs. On the other hand, bioactive nutraceutical peptides are likely to be cheap to manufacture and exhibit intrinsically low toxicity. Thus it may be possible to offset lower potency with a higher amount of peptide and PE.

### 4.6. Can Non-Ionic Surfactants be More Effective than Ionizable Surfactants?

Surfactant-based PEs have long been the leading candidates to improve oral absorption of poorly permeable actives. The most prominent surfactant categories include medium chain fatty acids (anionic), acylcarnitines (amphoteric), alkyl sulfates (anionic), and bile salts (anionic). A structurally diverse group of non-ionic surfactants have been shown to alter epithelial barrier integrity including fatty alcohol ethoxylates, ethoxylated sugar esters, alkylphenol ethoxylates, alkyl maltosides/glucosides, macrogol glycerides, and sucrose/lactose esters. Most non-ionic surfactants in this category are liquid or unctuous semi-solids at room temperature, which limits their formulation potential. Small quantities of liquid PE can be incorporated into solid dosage forms using adsorbent. However, it is more challenging to prepare powders that have acceptable properties for capsule filling or compaction into tablets (such as flowability, disintegration, dissolution, desorption). For example, solidification of Labrasol^®^ (Gattefosse, Saint Priest, Lyon, France) was achieved using relatively low quantities of silica (e.g., Neusilin^®^ US2, Fuji Chemicals, Nakaniikawa-gun, Toyama, Japan) (Figure 1). Tablets prepared using a Labrasol^®^ (50%), and Neusilin^®^ US2 had poor disintegration times, but this was corrected by inclusion of a disintegrant (Table 2). Use of additional excipients lowers the quantity by weight of PE that can be incorporated into the formulation. This approach is more therefore likely to be only useful for the most potent non-ionic surfactants (e.g., C_12_E_8_), where less PE is required to improve permeation.

Some non-ionic surfactants are solids at room temperature (e.g., sucrose and lactose esters) although a low melting point makes it challenging to incorporate into tablets. Growth in the number of poorly soluble drugs administered in lipid-based formulations (LBFs) has led investigators to assess delivery of macromolecules in non-aqueous vehicles in soft or hard gelatin capsules. An underpinning question is what are the advantages of non-ionic surfactants versus ionizable surfactants, salts of which are easier to formulate into solid dosage forms? In general, non-ionic surfactants are safer and more widely used as excipients and food additives, primarily as emulsifiers. Sucrose laurate is an excipient included in some marketed formulations, and, although it is present in low amounts, its status as an excipient reduces risks in development. In a recent head-to-head, sucrose laurate and Labrasol^®^ (Gattefosse, France) improved the flux of transmucosal marker molecules to a similar level to C_10_ and sodium undecylenate (C_11:1_) at comparable concentrations across isolated rat intestinal tissue mucosae [13]. Sucrose laurate caused less damage to isolated rat colonic mucosae compared with C_10_ at similar concentrations [63]. In intestinal instillations, sucrose laurate improved permeation of insulin in the absence of histological damage [64]. It is noteworthy that other sugar esters have recently been shown to exhibit permeation enhancement. For example, the enhancement action of lactose laurate was comparable to sucrose laurate [65]. Although sucrose laurate and other sucrose esters can be synthesized as monoesters and have demonstrated enhancement action in their pure forms, food and excipient grades are typically supplied as mixtures containing the soluble mono-ester assigned a hydrophilic lipophilic balance (HLB) of 15 and a significant proportion of insoluble di-, tri-, and polyesters (HLB < 5). Thus, dispersions formed by mixed sucrose esters are not simple micellar systems; they comprise mixed micelles and/or micro/nanoemulsions. There is a requirement for safe and selective methods for production and material separation.

Labrasol^®^ is another example of a complex blend of soluble and insoluble surfactants containing a large proportion of macrogol glycerides and 10% medium chain glycerides. Although Labrasol^®^ caused perturbation of isolated rat intestinal mucosae, there was a degree of separation between the enhancement action and the histology damage score [13]. The effect of the difference in dispersion properties has not yet been fully elucidated, although there is emerging evidence that the extent of the monomeric surfactant that is free to interact with the mucosal surface plays a role in permeation enhancement [66,67]. The effect of free surfactant on enhancement action is discussed further in Section 4.9.

### 4.7. Can Physical Hydrophobization Improve Passive Intestinal Flux?

The majority of PEs act by altering epithelial barrier integrity. In recent years, investigators have attempted to improve the lipophilicity of the active to facilitate passive intestinal permeation. Hydrophobisation can be achieved by chemical conjugation (prodrugs [68]) or physical complexation [9] (Section 2). Chemical modification is not ideal for large ionizable peptides as it is not practical to mask several amino acid side chains within a macromolecule. The reversible formation of a salt between the peptide and a complexing agent can lead to a more dramatic increase in lipophilicity. This occurs through a combination of charge neutralization, exposure of hydrophobic domains within the peptide and/or the introduction of lipophilic moieties via the counterion. The process of HIP, therefore, leads to extensive lowering of aqueous solubility, which can be addressed through encapsulation in nanoparticles (e.g., poly(lactic-co-glycolic acid [69]), solubilization in lipid-based formulations [70], and non-aqueous solvents [22,71]. Incorporation of HIP complexes in the non-aqueous vehicle may prevent enzymatic degradation, limit pH dependent complex destabilization, and counterions may also perform as PEs [72]. Given the successful application of lipoidal vehicles in oral delivery of cyclosporin, there is research effort assessing the factors that impact loading in LBFs [70]. There are cases where HIP does not result in loss of aqueous solubility [73] and do not immediately breakdown at high pH values where deprotonation of cationic functional groups is known to occur [74]. However, further investigation is required to understand the permeability of soluble HIP complexes. Other considerations require kinetic and thermodynamic assessment of the dissociation of the hydrophobized peptide complex during permeation.

### 4.8. Mode of Action Studies are Required to Provide Evidence for a Paracellular Effect

The mechanism by which PEs alter the intestinal barrier has implications for safety and approval of PE-containing oral macromolecule formulations. In general, few PEs solely act via a paracellular mode of action, and it is necessary to distinguish paracellular and transcellular pathways across intestinal epithelia. Reduction in TEER for example or an increase in flux of paracellular transport markers is not direct proof of a paracellular mode of action, as transcellular perturbation can also reduce TEER and increase such fluxes. The absence of cytotoxicity in common cell viability assays (e.g., MTT) at concentrations that cause alteration to TEER and marker transport favour an interpretation relating to paracellular pathways. However, cytotoxicity assays based on mitochondrial enzymes may not represent the first sign of membrane perturbation. In this case, it may be appropriate to combine the use of MTT assay and techniques that directly evaluate membrane perturbation (e.g., lactate dehydrogenase (LDH) release assay). Two-path impedance spectroscopy has been used to show that C_10_ acts via a paracellular mode of action in vitro at low concentrations [75]. However, this method requires strict verification that the PE does not cause intracellular uptake of a paracellular dye [76], and applications that do not assess membrane integrity can overestimate the contribution of the paracellular route [75]. A recent study showed restoration of barrier integrity and a parallel decrease in absorption of a model peptide following cessation of intra-duodenal perfusion of C_10_ in rats [77]. This reversible action was equated to a paracellular mode of action, the rationale being that, had the effect been related to mucosal perturbation, the absorption of peptide would have continued to remain high. This conclusion does not, however, take into account the capacity of the GI tract to undergo rapid repair following perturbation [78]. Overall, more direct mode of action studies are required to confirm a paracellular mode of action. 

### 4.9. Growing Need for Simulated Intestinal Fluid in PE Experiments

Over the last 30 years, there has been extensive effort to predict the behaviour of oral formulations in humans from solubility and release characteristics in vitro. This has given rise to important topics in biopharmaceutics including bioequivalence testing and in vitro/in vivo correlations. Solubility enhancement strategies highlighted the importance of replicating in vivo fasted and fed state conditions in order to predict release characteristics in humans, however there has not been the same emphasis on how constituents of intestinal luminal fluids affect intestinal permeability and in vivo absorption in preclinical models (including cell culture monolayers grown on Transwell^®^ supports, isolated tissues mounted in Ussing chambers, everted and non-everted intestinal sacs, open/closed loop instillations, tablet insertion into gut loop, single-pass intestinal perfusion, infusion via intubation, oral administration as liquid or solid dosage forms). 

Gastrointestinal fluid contains a complex mixture of endogenous secretions (including gastric, pancreatic, luminal, and biliary secretions, sloughed cells) and dietary substances (nutrients, drugs, microorganisms). This heterogeneous milieu consists of a cocktail of ions, bicarbonate, enzymes, mucin, bile acids, phospholipids, carbohydrates, proteins, amino acids, lipids, fatty acids, indigestible solid particles, and lysates from sloughed epithelial cells and non-viable microorganisms. In principle, the luminal milieu can modulate enhancement action of PEs. In the simplest example, PEs that are peptides/proteins may be degraded by proteolytic enzymes. Not all protein-based PEs are inactivated by luminal fluid as evident from the fact that many protein-based toxins stimulate electrogenic chloride secretion and TJ openings [79]. It remains to be seen whether peptide-based PEs can consistently modulate epithelial permeability or whether these additives must be chemically-modified to increase stability (e.g., cyclisation, synthesis of all D-forms, amino acid substitutions) or mixed with excipients that prevent enzymatic degradation of the therapeutic peptide (e.g., citric acid, soybean trypsin inhibitor). Other structural PEs categories can also be enzymatically degraded in the small intestine. Macrogol glycerides (e.g., Labrasol^®^, Gattefosse, France) are substrates of digestive lipase [80], although the degradation products in this instance are free medium chain fatty acids which may contribute to permeation enhancement [81]. 

It is not only chemical degradation that can attenuate the enhancement action. PEs that have demonstrated the most significant effect on epithelial permeability are soluble surfactants. This surfactant type exists in the monomeric form up to threshold concentration above which they form micelles. This critical micelle concentration (CMC) is both a measure of the solubility of the monomeric surfactant form and a direct measure of when surfactants begin to form micelles. Ionic surfactants with high CMC values are generally good detergents because it is the monomeric form of the surfactant that is responsible. As detergent-like membrane perturbation caused by surfactants is largely driven by the monomeric form of the surfactant, any physiological factor that reduces the CMC can potentially reduce efficacy. The CMC of ionizable surfactants such as medium chain fatty acids, acylcarnitines, and alkyl sulfates can be decreased by increasing the ionic strength of the medium, as the addition of counterions reduces repulsion between anionic hydrophilic head groups, which makes micellization favorable at low concentrations [82]. Recently, enhancement action of C_10_ and sodium dodecyl sulphate (SDS) was increased in hypotonic conditions (achieved with NaCl) in a single-pass intestinal perfusion in rats [83], an effect that could relate to a reduction in the CMC. In the case of ionizable surfactants that contain weakly acidic or weakly basic hydrophilic head groups, alteration to pH can transform a surfactant from its soluble form to its insoluble form, which although still capable of lowering surface tension, has reduced capacity to act as a detergent.

Divalent cations such as calcium found in milk can decrease the permeation enhancement of anionic surfactants through precipitation of inactive salts [84]. On the other hand, alteration to pH may increase permeation enhancement. For example, reduction in the regional pH with an amphoteric surfactant may increase the proportion of the surfactant in the cationic form, which is likely to have a greater affect. Additionally, it has been proposed that co-solvents that increase the solubility of insoluble surfactants can improve aqueous solubility and enable greater interaction with the cell membrane [66]. Several other factors may modulate the free surfactant concentration within the small intestine. These include adsorption to undigested solid particles and oil droplets and incorporation into colloidal structures (e.g., mixed micelles with free fatty acids and bile salts). Depending on the quantity and type of materials within the GI tract, free surfactant monomers may be efficiently replenished from micelles, but as the window for efficient enhancement can be shortened by dilution, spreading, and absorption of the PE itself, the availability of surfactant within that window may be quickly diluted. Phospholipid and bile salt constituents of simulated intestinal media can attenuate the effects of alkyl maltopyranosides through the formation of mixed micelles [67]. Fasted state simulated intestinal media (FaSSIF) containing taurocholate, phosphatidylcholine in buffered isosmotic buffer salt solution (pH 7.1) also attenuated the permeation enhancement of palmitoylcarnitine and hexyphosphocholine surfactants in Caco-2 monolayers [85]. In the same study, FaSSIF did not affect the enhancement action of non-surfactant PEs, EDTA and 3-nitrocoumarin. In some cases, mixing surfactants can increase enhancement action still further [86]. Combinations of medium chain fatty acids with PEG-8 glycerides increased permeation of FD4 across isolated rat intestinal tissue by 10-fold over the respective individual agents [87].

A recent study assessed the effect of FaSSIF and fed state simulated intestinal fluid (FeSSIF on the absorption of BCS Class I, II and III drugs in a single pass intestinal perfusion (Roos et al., submitted). Even in the absence of PEs, there was a significant difference in the absorption of atenolol, enalaprilat, ketoprofen, and metoprolol between FaSSIF and FeSSIF. This result was not surprising as an analysis of 92 clinical datasets found that 67% of BCS Class I drugs had no food effect, 71% of Class II had a positive food effect, and 61% of BCS Class III had an adverse food effect [88]. Additionally, there is evidence of reduced drug absorption in rats when metoprolol was perfused in aspirated fed state intestinal fluids compared to fasted state aspirates [89]. Absorption of selected PE active combinations were reduced in FeSSIF, which the authors attributed to the presence of colloidal structures formed by lecithin and bile acids in FeSSIF (Roos et al., submitted). Surfactant PEs have potential to emulsify dietary lipids and can form mixed micelles with both soluble (e.g., bile salts, ionized free fatty acids) and insoluble surfactant (glycerides, phospholipids), and as the CMC of these mixed micelles is typically lower than the native surfactant, these structures are capable of lowering the sufficient quantity of free monomeric surfactant that is available to perturb membranes. A conclusion is that luminal surfactants can exert a protective effect against mucosal perturbation by ionizable surfactants. It is noteworthy that the presence of a luminal surfactant does not change the CMC of the PE, so if there is a large excess of PE over the luminal surfactants, it will exist in the molecular form at its CMC and will therefore be available to exhibit transcellular perturbation. 

Further work is required to determine the effect of luminal composition on the absorption of soluble macromolecules and the enhancement action of PEs. A question arising from studies showing the effect of luminal constituents on PE actions is what constitutes the typical composition within the GI tract? Given the variability in free fatty acid, lipids, and bile salts in human intestinal fluid [90], it may be difficult to precisely identify the type and amount of substances that must be included in the SIF for permeability studies. Additionally, it remains unclear whether there is a substantial difference in the concentration and type of constituents in bulk luminal fluid and local extrinsic mucus gel layer. Therefore, it is also unclear if SIF for permeability testing should represent bulk luminal fluid (and hence mirror dissolution media), or whether it should mirror the composition of the extrinsic mucus gel layer. In addition, it is difficult to use SIFs designed for dissolution testing [91] in cell- or tissue-based in vitro assays, as several constituents of intestinal fluid (e.g., bile acids, phospholipids, and free fatty acids) are themselves capable of altering permeability and causing local perturbation. There have been efforts to develop biorelevant media that do not alter the viability or barrier integrity of Caco-2 monolayers over short periods [92]. By necessity there are reductions in the concentrations of bile salts in the SIF used in vitro compared to intestinal aspirates (e.g., free fatty acids). Other investigators have opted to use full biorelevant media and overcome damage to Caco-2 monolayers using overlying biosimilar mucus [93,94]. It may be that studies with SIF should begin with more robust in situ models such as the rat single pass intestinal perfusion [83,95,96]. More research assessing the effect of PEs in SIF would also help investigators understand how a growing number of lipoidal dispersions improve intestinal permeability (e.g., TPE^TM^, Chiasma, Ness Ziona, Israel [17]).

### 4.10. Improving PE Action in the Dynamic GI Tract

Of those PE that have been assessed in oral formulation of macromolecules in clinical trials, only a modest single digit increase in oral BA has ever been observed. The majority of PEs are effective in static delivery models, where the PE and active are co-delivered in liquid dosage forms to cells, isolated tissues or tied intestinal loops for extended periods. In these models, the PE and active are typically presented to the epithelial surface above a threshold concentration for an extended period of several hours. This provides the PE sufficient time to alter barrier integrity in the presence of a high concentration gradient of both macromolecule and PE. These optimal conditions do not prevail in the human GI tract following oral delivery of a solid dosage form (Figure 2).

Few published studies assess the effect of PEs in oral dosage forms. The majority of oral peptide dosage forms in clinical development are enteric coated, and so that active and PE are co-released in the small intestine. Relatively quick transit along the small intestine attenuates enhancement in the GI lumen as the PE requires time at a focal point within the lumen to alter integrity and facilitate permeation of the macromolecule, while not being too rapidly absorbed itself (as is the case with C_10_ and SNAC). This is problematic for all PEs, in particular, those that do not cause a rapid decrease in barrier integrity. Investigators have shown that confining PE and macromolecule over a focussed area [97] and the synchronous co-release of both in high concentration in a short period [78,98,99] can improve enhancement. The fluid volume within the stomach and small intestine is another potential variable. Intestinal fluid is not homogeneously spread across the small intestine, it forms fluid-filled pockets of varying number and volume [100], which may give rise to variable absorption. This is because dissolution of PE in a small volume will lead to greater enhancement action than if the PE is dissolved in a larger fluid volume. As there has been little research assessing the effect of release metrics on permeability, it follows that few delivery technologies have been developed to improve localization of PE and active at the intestinal epithelium. Initial experimental strategies tested include mucoadhesives, superporous hydrogels [101], electronic capsules (e.g., IntelliCap^TM^ [102]), controlled release of PE [98,103,104], intestinal patches (Section 4.11) and pharmacological motility (e.g., loperamide). Overall, the goal of optimizing co-presentation aims to create a diffusion gradient to improve the rate and extent of enhancement action and all PEs to reach and sustain a threshold concentration required to improve intestinal flux.

Dissolution of the PE and active is another factor that must be considered for effective translation. Peptides often exhibit their lowest solubility at the isoelectric point (pI) and greatest solubility at pH values below the pI owing to ionization of basic side chains in acidic conditions. Insulin (pI 5.5) for example, dissolves well in acidic media, but has low solubility at neutral pH. Peptides with low potency and low intrinsic solubility may dissolve more slowly in the small intestine, which may lower the concentration gradient that drives flux. Gradual dissolution may prevent a PE from reaching and sustaining a threshold local concentration for enhancement (e.g., chitosan [105], medium chain fatty acids [106]). Investigators have attempted to improve release through the inclusion of disintegrants [97], or via formulation as liquids [46].

### 4.11. Intestinal Patches to Co-Localise PE and Active

Intestinal patches are a more recent strategy for localization of poorly permeable solutes at the small intestinal epithelium. Some of the earliest patches did not include PEs [107], but their inclusion in more recent iterations suggests they may be required for optimal patch performance. Assessment of a 13 mm patches (coated on one side with ethyl cellulose backing) in Caco-2 monolayers showed a modest two-fold enhancement of insulin and exenatide flux [108]. In the same study, there was significant absorption of insulin in patches, especially those that contained the PE, dimethyl palmitoyl ammonio propanesulfonate (PPS) (0.5% w/v). Other PEs incorporated into patches include SDS, polyoxyethylene hydrogenated castor oil 60 [109], thiolated polycarbophils [110] and Labrasol^®^ [111]. Patches are delivered in enteric-coated capsules, so there are no compaction forces associated with tableting. More recently, patches have been miniaturized into micropatch formats, which offer a large surface area for absorption. Oral delivery of capsules containing micropatches (50 U/kg insulin and 0.2 mg PPS) admixed with citric acid (15 mg) decreased plasma glucose in rats by 22% compared to a drop of 54% with free insulin (1 U/kg, s.c.) [112]. Further optimization is required to ensure adequate adhesion within the GI tract and optimal retention at the mucosal surface. 

### 4.12. Is Safety of PEs a Real Impediment to Translation?

Safety is a widely held concern to the application of PEs in oral delivery (reviewed in [113]). This concern can be sub-categorized into those related to individual or sub-classes of compounds and concerns relating to modulation of intestinal barrier integrity on a chronic basis. Over 250 compounds have been shown to alter intestinal permeability, but only about a dozen of these are realistic candidate PEs for clinical trials. The reversible modulation of TJ opening might be viewed as a safer approach to improving permeation relative to transcellular membrane perturbation. 

The first-generation TJ modulators acted via intracellular cell signaling pathways which alter TJ architecture from within epithelial cells. There are risks related to PEs that alter permeability through ubiquitous cell signaling processes such as protein kinase C [114]. Although additional pharmacological effects observed with currently ‘allowed’ excipients, this is secondary to a primary action, examples including depression of the CNS (e.g., ethyl alcohol), inhibition of metabolic enzymes and transporters (D-alpha-tocopherol polyethylene glycol 1000 succinate [115], and Cremophor EL [116]) and interference with metabolic enzymes [116]. Most second-generation paracellular PEs selectively disrupt the interaction between TJ proteins in adjacent cells [8], although the consequences of modulating TJ at other barriers is not yet entirely clear [9]. 

The alternative is to use surfactant-based PEs that increase permeation via transcellular perturbation. High concentrations of these additives often cause temporary mild local mucosal injury, but when diluted are less likely to reach the high concentrations required for cell perturbation at other sites within the body even if absorbed. Additionally, some leading candidates in this category have food additive (e.g., C_10_) or GRAS status (e.g., SNAC) or are pharmaceutical excipients (e.g., sucrose esters). There may be a relatively low risk of systemic toxicity for such surfactants, although there are ongoing concerns related to local mucosal perturbation. The majority of evidence to date suggests that surfactant PEs do not possess a discrete mechanism of action; instead they act by directly compromising the integrity of enterocyte plasma membranes. This category of PE has been assessed extensively in clinical trials over the last 20 years, and while there have been no major adverse events reported, there has also been only a modest effect on oral BA in most trials. It remains to be seen if new technologies designed to improve residence of PE and active at a focal point within the intestinal lumen may result in more histological damage. It is difficult to determine if PEs cause histological damage at the site of release in the human GI tract, owing to difficulty pinpointing the site of release and performing a biopsy within the enhancement window and before epithelial repair. In one of the few clinical trials assessing the effect of PEs on histology score within the GI tract, administration of Doktacillin^TM^ suppositories (ampicillin (250 mg), C_10_ (25 mg) and hard fat (950 mg, Pharmasol^TM^ B-105, NOF Corp. Shibuya-ku, Tokyo, Japan)) improved rectal BA of ampicillin from 13 to 23% [117]. Doktacillin^TM^ suppositories caused a significant increase in the average histology score from 0.62 in the control group (before administration) to a score of 1.94 after administration of the suppository (25 min). The average histology score measured three hours after administration of Doktacillin^TM^ was 0.96, suggesting the barrier can repair from mucosal perturbation. The authors note that C_10_ itself and hyperosmolarity of the rectal fluid contribute to the histological damage score. Irrespective of whether C_10_ causes hyperosmolarity, exposure time in the rectum is longer than any equivalent segment of the small intestine owing to longer residence time. There is also a higher concentration of PE due to a lower amount of rectal fluid than in other intestinal segments. To allay fears over chronic daily use of PEs, some may be confined to administrations with a different dosing regimen. Some PE formulations tested in trials include for example, a once-weekly tablet zoledronic acid with C_10_ (Orazol^®^, Merrion Pharma, Ireland [118]), which would provide time for recovery of barrier integrity. Nonetheless, time to recovery from PEs that cause mild mucosal perturbation does not seem to be a problem in clinical trials so far (e.g., daily administration of semaglutide with SNAC [28]). To date, there has been no evidence of intestinal problems that are unrelated to the GLP-1 class in any oral semaglutide trials performed.

It is tempting to justify the use of PEs that cause mild reversible mucosal damage by citing precedence for the use of substances that cause GI disturbances. Regarding medicines approval, it is not possible to justify the use of PEs that act to improve flux via transcellular perturbation by citing the side effects of drugs, excipients or food additives. However, a PE may not be listed as such in any given formulation, and therefore their use may be more comparable to other additives that have been shown to cause membrane perturbation in vitro (e.g., surfactants [119]). Comparison of PEs to established excipients does help to show that mild mucosal perturbation may be common. 

There is concern that altering the integrity of the intestinal epithelium, irrespective of mode of action, may facilitate colonization of epithelial cells and invasion of deeper tissues by opportunistic pathogens. While there is no evidence to date that oral PE-macromolecule dosage forms cause infections in clinical trials, it is important to mention that if this is true risk, then potential clinical manifestations range from asymptomatic infection to diarrhea, to more serious infections depending on the micro-organism and host genetic variability [120]. This concern would seem to be more plausible for PEs that cause reversible mucosal damage, and not TJ modulators, because the maximum opening of the TJ is far smaller than the diameter of a bacterium [121]. Compromising the integrity of the intestinal mucosae is more likely to facilitate the translation of microorganisms. In Caco-2 monolayers, co-incubation of octyl phenol ethoxylate (Triton X-100) with *E. coli* increased translocation, although whether such translocation occurs in vivo has not been assessed [118].

There is the additional concern that other luminal bystanders may be translocated within the enhancement window. There is evidence that ischemia can increase levels of plasma lipopolysaccharide (LPS) [122] which may contribute to septic shock [123]. Given that PEs cause only a modest uptake of small peptides co-localised in the formulation, it seems unlikely that large luminal xenobiotics (such as LPS or LPS fragments) that are spread diffusely across the GI lumen would be appreciably absorbed. Again, this potential issue may be less likely for PEs that act via the opening of TJs, as the diameter of an open junction within the small intestine is estimated to be 10 nm [84]. It is noteworthy however, that altered expression of the TJ protein claudin is associated with intestinal bowel disorders [124]. The surfactants hexadecylphosphocholine and octylphenol ethoxylate increase permeation of LPS across Caco-2 monolayers [125]. Co-delivery of C_10_ (154 mM) or penetratin (2 mM) with LPS to mice for seven consecutive days did not cause LPS induced hepatic damage, as measured by plasma levels of alanine aminotransferase (ALT) and aspartate aminotransferase (AST) which are both released during hepatic necrosis [126]. A significant elevation in those enzymes was observed when LPS was co-delivered with taurodeoxycholate (96 mM) in the same study. However, as there was no control treatment in the absence of LPS it remains unclear whether elevated levels of hepatic necrosis markers relate to LPS or PE only. There is also evidence from a controversial study that prolonged exposure of two common emulsifiers (polysorbate 80 and carboxymethyl cellulose) causes low-grade inflammation in healthy mice and colitis in predisposed mice via altering the microbiome [127]. It is not known if these findings have any relevance for human exposure to excipients.

### 4.13. Convergence between Delivery Concepts, Intestinal Physiology, and Formulation Science

In order to address the hurdles to translation, there must be convergence between delivery concepts, intestinal physiology, and formulation. The material and physicochemical properties of the PE must be considered more; especially as PEs can be a significant proportion of the formulations. As noted in Section 4.6, it can be challenging to incorporate liquid or semi-solid PEs into oral dosage forms. Salts of ionizable surfactants are more readily incorporated into tablets. However, there remains the likely requirement for other excipients in the formulation, which in some instances may require a reduction in the quantity of PE to accommodate these additives. The type of formulation and process additives and their quantity in the final manufacturable scalable dosage form are important considerations as delivery researchers begin to assess optimal luminal presentation, be it for immediate release or controlled release. There are also practical considerations in the progression of PEs to clinical testing, such as availability of GMP grade material or a specification. There is the requirement for extensive information in any regulatory submission, including details of manufacture and purification procedures, extensive physical and chemical properties of any new PE and provision of supporting safety data (non-clinical/clinical).

Reliable animal models are prerequisite for identification of promising macromolecule drug delivery systems that have potential to translate to humans. There is variability between key anatomical and physiological parameters in humans and those observed in rats, pigs and dogs (Table 3). These include differences in gastric emptying time, gastric retention time, absolute water content, transit time, local pH, gut length, mucous thickness, and average stomach capacity [128,129]. It is possible to purchase minicapsules and manufacture minitablets for oral delivery to rats, but there are additional challenges. In the case of minicapsules it can be difficult to load sufficient levels of PE into the capsule shell without prior granulation (e.g., by roll compaction) and these formulations are difficult to enteric coat by manual dip coating (thus necessitating the use of specialist coating equipment). It is possible to purchase single- or multi-tip tooling for the preparation of minitablets, although this requires specialist tableting equipment (granulator, spray coater, and tablet press) and expertise in formulation and process development. There are very few studies where complete dosage forms are administered to rodents, and therefore the formulation factors that impact translation are rarely assessed in this widely used animal model. There is the argument that formulations administered to rats have little relevance to formats tested in larger animals and humans, and there is therefore a need for studies in more representative animals and humans. Nevertheless, if the impediments to translation of PE dosage forms can be partially optimized in rodent models, this could better inform development of formulations in humans. Oral peptide research in rodents typically involves oral gavage of liquid mixtures and not dosage forms tested in humans. This is the same for rat in situ static intestinal instillations, which help identify PEs in ad-mixtures with payloads, but do not offer information on transit time, fluid volume, pH conditions within the small intestine nor do they adequately model dissolution of PE dosage forms. Hence, translation of results from rats to humans is limited by fundamental anatomical and physiological differences in intestinal length (1.8 m versus 5.9 m [128]), transit time in the fed (20 h versus 4 h [130]) and water volume (0.06 g/cm gut length versus 0.58 g/cm [130]).

Dosage forms of similar dimensions to those used in humans are amenable to administration in dogs and pigs. However, as there are longer gastric emptying times in these animals, immediate release PE dosage forms intended to increase flux within the stomach (e.g., semaglutide) could be more effective in dogs and pigs owing to greater residence time (localization of PE and active), although such data may not be replicated in humans due to shorter residence times. In the case of enteric coated macromolecule formulations, longer gastric emptying times in pigs and dogs may result in a longer delay to onset of drug absorption compared to humans. The pH in the stomach of healthy fasted dogs is highly variable (pH 1–8) [129,131], leading some investigators to pre-administer an acidic solution containing 0.1 M HCl and 0.1 M KCl [132] or the hormone pentagastrin to stimulate endogenous acid secretion [133]. Both approaches aim to standardize the pH prior to oral administration to animals. In the absence of pH adjustment, the stomach of dogs may not adequately model release of enteric-coated oral peptide formulations. In the small intestine, transit time is shorter by half in dogs compared to humans and pigs. Faster transit reduces the absorption window for drugs, which gives rise to differences in oral BA. As the importance of local GI retention of PE and macromolecule is requisite for efficient enhancement, study of PEs in dogs may underestimate the action of these additives. Further interspecies divergence is observed at the intestinal epithelium, where mucous thickness is appreciably larger in dogs compared to humans. The anatomy physiology and diet of pigs closely aligns to humans [134], and in some aspects (but not all aspects, e.g., gastric residence time) pigs are considered an appropriate model to assess GI permeability [133]. The composition of enterocyte membranes in different animal models [128] could also play a role in different susceptibility to surfactant PEs [9]. It is also worth emphasizing that there can be differences in GI physiology in different disease states [135], which could also influence the action of PEs.

Interspecies differences highlight that the best model of human is human [136], nevertheless combinations of animal models are still relevant for optimization of oral PE macromolecule formulations and ultimately translation into effective dosage forms in man. Although most oral peptide formulations fail to progress to clinical evaluation, poor translation between animal models and humans is likely to be one of many reasons. There is nearly always higher BA in open- and closed-loop instillations and single pass intestinal perfusion compared to oral dosage forms, but these models still provide a necessary first hurdle with a relatively low bar; it is then the challenge for formulation and delivery researchers to develop technologies that recreate such presentation in appropriate oral dosage forms.

## 5. Conclusions

Discovery of safe, effective, and formulation-compatible PEs for oral delivery of macromolecules has been a priority for investigators. Research on PEs has branched into the discovery of agents that target opening of TJs [47] or using substances with established use in humans [106]. Recent studies have examined (i) the behavior of PEs and acidifiers in the intestinal lumen, (ii) factors that impact enhancement within the dynamic environment in the GI tract in vivo (e.g., intestinal fluid, tonicity, exposure time), (iii) physiochemical properties of PEs that give rise to enhancement, (iv) effect of PE combinations, and (v) learnings from clinical investigations [140,141]. These studies have highlighted the need for greater emphasis on the hurdles to translation, in particular, development of concepts to optimize the co-presentation of PE and macromolecule payload in high concentration for as long as possible at the intestinal wall. Greater focus on this area may improve the likelihood of translation. 

## Figures and Tables

**Figure 1 pharmaceutics-11-00041-f001:**
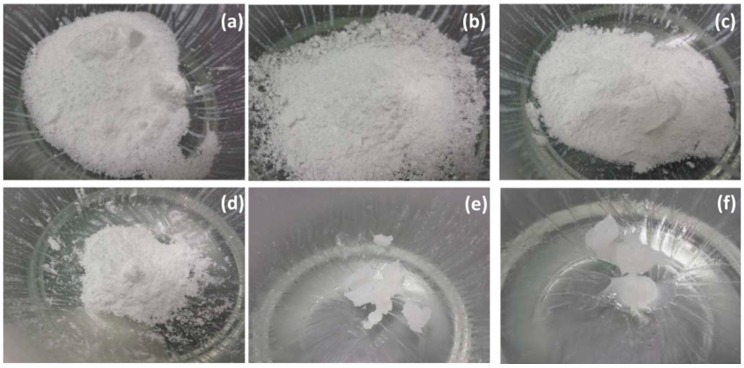
Visual effect of Neusilin^®^ US2 on physical state of Labrasol^®^. Ratio of Neusilin^®^ US2 to Labrasol^®^ are (**a**) 1:0, (**b**) 0.75:0.25, (**c**) 0.67:0.33, (**d**) 0.5:0.5, (**e**) 0.33:0.67, (**f**) 0.25:0.75 (Maher unpublished).

**Figure 2 pharmaceutics-11-00041-f002:**
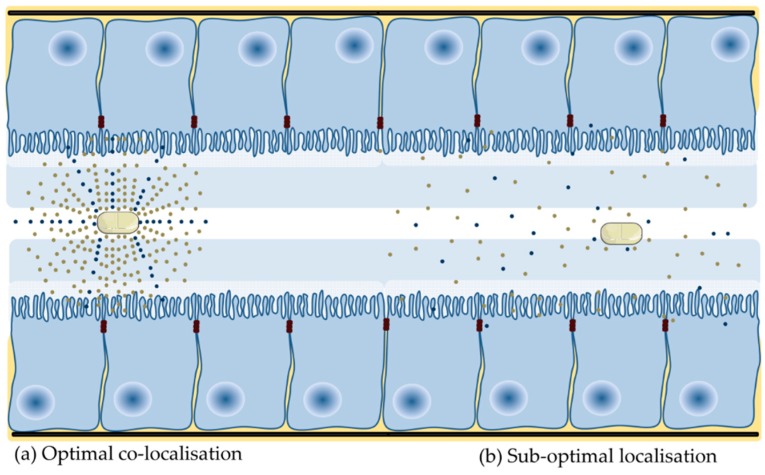
Presentation of PE and active at the small intestinal mucosa. (**a**) Co-localization of PE and active at the intestinal epithelium ensures that the PE is present at a high concentration and is present for long enough to alter barrier integrity. (**b**) A high local concentration of active provides the drive force for intestinal flux. A low release rate and more gradual dissolution of PE dosage forms, fast transit, spreading and dilution in luminal fluid and interaction with constituents of luminal fluid will impede optimal co-localization because the concentration of both are ultimately not high enough.

**Table 1 pharmaceutics-11-00041-t001:** Properties of selected approved peptides and their routes of delivery (source of information: Health Products Regulatory Authority (HPRA) summary of product characteristics (SPC), Drugbank, PubChem, and Welcome Compound Report Card).

Active	Mw	Dose	Frequency	Route	t½	LogP ^†^	BCS	Oral BA
Desmopressin	1069 Da	1–4 mcg	Daily	sc	~2.8 h	−4	III	0.17%
Octreotide	1019 Da	200 mcg	Thrice daily	sc	~1.7 h	−1.4	—	Phase 3
Cyclosporin	1203 Da	280 mg	Daily	iv inf.	~8.4 h	7.5	II	27%
Vancomycin	1449 Da	1500 mg	Twice daily	iv inf.	~7.2 h	−2.6	III	Local
Salmon calcitonin	3432 Da	16.7 mcg	Daily	sc	~1.3 h	−16.6	—	Phase 3
Semaglutide	4114 Da	500 mcg *	Weekly	sc	~168 h	−5.8	—	Phase 3
Exenatide	4186 Da	10 mcg	Daily	sc	~2.4 h	−21	—	Phase 1
Insulin degludec	6108 Da	350 mcg	Daily	sc	~25 h	−4.9	—	—
Insulin aspart	5832 Da	1.8 mg **	—	sc	~1.4 h	—	—	—

* oral dose in clinical testing [28]; ** estimated daily dose; ^†^ estimated logP (XLogP3-AA [40]); BCS: Biopharmaceutics classification system; t½: plasma half-life; LogP: octanol water partition coefficient.

**Table 2 pharmaceutics-11-00041-t002:** Disintegration times and break strength values for a panel of formulations containing Labrasol^®^, Neusilin^®^ US2 and a disintegrant (Croscarmellose Sodium).

Formulation Additives	Disintegrant (% w/w)	Tableting Pressure (psi)	Disintegration Time (min)	Break Strength (N)
Labrasol and Neusilin^®^ US2 (1:1)	0	1000	>60	29.1 ± 2.9
Labrasol and Neusilin^®^ US2 (1:1)	0	2000	>60	68.8 ± 3.2
Labrasol and Neusilin^®^ US2 (1:1)	5	1000	5.5 ± 0.2	49.8 ± 6.2
Labrasol and Neusilin^®^ US2 (1:1)	5	2000	4.9 ± 0.3	72.4 ± 2.7

**Table 3 pharmaceutics-11-00041-t003:** Potential effects of GI physiology in different animals on the action of PEs.

Anatomical/Physiological Property	Species	Influence on PE Action
Gastric emptying time (h)	Human: 1 h [130] Rat: 0.7–2.1 h [130] Dog: 3.9–5.3 h [130] Pig: 1.5–6 h [130]	For immediate release dosage forms, slower gastric emptying in pig and dog than in humans may increase gastric residence time of PE and payload, thus overestimating enhancement. For enteric dosage forms, slower gastric emptying, may delay dissolution in the GI tract and ultimately increase Tmax in these species versus humans.
Gastric fluid volume (mL)	Human: 118 mL [137] Rat: 2.29 [137] Dog: 500–1000 mL [137] Pig: 278 mL [137]	For immediate release dosage forms, the larger volume in dogs may result in greater dilution of PE to below a threshold for enhancement action, thereby underestimating enhancement.
Stomach pH	Human: 1.7 [128] Rat: 3.9 [138] Dog: 1.5 [128] Pig: 1.7 [128]	As many PEs that have progressed to clinical testing in oral formulations are weak acid surfactants, differences in solubility can be observed if there is variation in gastric pH. This gives rise to differences in enhancement as acidic surfactants are more effective in their ionizable form at high pH.
Small intestine transit time (Fasted state) (time (h) and length (m))	Human: 3–4 h [139] Human: 6.25 m [137] Rat: 4–5 h [128] Rat: 0.34 m [137] Dog: 1.5 h [137] Dog: 2.48 m [137] Pig: 3–4 h [137] Pig: 14.2 m [137]	Faster transit may reduce the exposure of PE and payload at the epithelium, thereby reducing enhancement, and potentially underestimating the effects of the PE. A short transit time does not strictly mean faster movement, as length of the small intestine is different in different species.
Small intestine fluid volume (total and g/cm)	Human: 212 mL [137] Human: 0.6 g/cm [130] Rat: 3.9 mL [137] Rat: 0.06 g/cm [130] Dog: 300 mL [137] Dog: 0.9 [130] Pig: 476 mL [137] Pig: 0.62 [130]	Differences in fluid volume, or more specifically the volume and number of intestinal fluid pockets in the small intestine could lead to differences in the regional concentration of PE and payload, as well as differences in dissolution rate. This could lead to under- or overestimation of enhancement.
Duodenal mucus thickness (µm)	Human: 15.5 µm [137] Rat: 30.6 µm [137] Dog: — Pig: 25.6 µm [137]	Difference in the thickness of the protective mucus gel layer overlying the epithelium has potential to modulate enhancement.
Small intestine diameter	Human: 5 cm [137] Rat: 2.5–3 mm [137] Dog: — Pig: —	The diameter of the intestinal lumen may impact the proximity of enteric formulations to the epithelium and ultimately impact co-localization of PE and payload.
Plasma membrane phospholipid composition of intestinal epithelium	Human: — Rat: — Dog: — Pig: —	There are differences in phospholipid composition in different species [128], which may impact sensitivity to perturbation by surfactant PEs

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
