# Peer review of "Application of Permeation Enhancers in Oral Delivery of Macromolecules: An Update"

_pharmaceutics, 2019, doi:10.3390/pharmaceutics11010041_

Reviewer 1 Report

The manuscript by Maher S. et al. entitled "Application of permeation enhancers in oral delivery of macromolecules: an update" is a review of the field of research devoted to the study and development of permeation enhancers (PEs) and oral formulations suited to improve transport of poorly absorbed pharmaceuticals across the intestinal epithelium. As indicated by its title, the review focuses on the current state of the field, including PEs in early development and some of the challenges that continue to exist in designing safe and effective formulations for future clinical use.

Evaluation: Collectively, the authors possess great amounts of expertise as researchers within this active field of pharmacology, and they frequently review the continuing progress that occurs. In this context, the present review provides a useful account the current state of the field, and no doubt it will be of interest to other researchers working within this area. In my opinion, it is well organized and written, and I have no critical comments to offer. The only detail I question is the informative value of Figure 2: What exactly is conveyed by the graphics (apart from a pill speeding down the intestine)? Generally, I fully support the inclusion of graphic art in review articles to liven-up the text, but this particular figure leaves me puzzled.       

Author Response

Question 1: The only detail I question is the informative value of Figure 2: What exactly is conveyed by the graphics (apart from a pill speeding down the intestine)? Generally, I fully support the inclusion of graphic art in review articles to liven-up the text, but this particular figure leaves me puzzled.

Author response: We have amended this figure and legend to emphasize the importance of co-localization of PE and active at the intestinal wall.

Reviewer 2 Report

This is a very interesting review covering very many practical challenges and problems to the application of permeation enhancers to oral delivery of macromolecules. Although some of the authors have recently (2016) published a similar review, some important recent advances in the field warrant an update of the state of the art. I have only minor suggestions.

Some passages or words are unclear:

Line 46 What do you mean by: Technology………may “diversity” the type of compounds….?

Line 50 “… and then rely on the delivery system to address sub-optimal PK characteristics after the fact,  or whether…….” This sentence is too long and not very clear . Please rephrase it for better comprehension.

Line 87: What do you mean by “reductionist models”, In vitro cell culture models or others ? It would help to specify “reductionist models, such as………..”

Line 93: is there a reference ?

Line 205. “certainly” ??

Line 215: “More recently, a structure activity assessment  of PIP-640 showed that residues….” 

Line 419-20: “the permeation enhancement of anionic surfactants”

Line 447. …was perfused rather…. than "perfusion"

Line 549: “principle” do you mean “primary” ?

Lines 579-80:   “due to a lower of intestinal fluid for dilution.”  Better would be:.... due to a lower amount of rectal fluid than in other intestinal segments.

There are some inconsistencies in the use abbreviations:

In Table 1: substitute sCT with salmon Calcitonin; use twice daily or thrice daily in place of the abbreviations bd and tds.

Line 214: FD4 First time should be spelt out as Fluorescein isothiocyanate-dextran and MW range

Line 244 use abbreviation sCT

Line 342: what does PLGA stand for?

Line 462: here I think you mean “human intestinal fluid” not SIF

Line 466: use abbreviation SIF

Some commas are missing:

line 349: However,

line 370: the rationale being that, had the effect….., the absorption….

Line 467: …mirror dissolution media), or whether...

Line 583: Nonetheless, time…

Line 612: ….noteworthy  however, that…..

Line 662: Hence, translation….

Author Response

REVIEWER 2

Question 2: Line 46 What do you mean by: Technology………may “diversity” the type of compounds….?

Author response: Amended to “Delivery systems”

Question 3: Line 50 “… and then rely on the delivery system to address sub-optimal PK characteristics after the fact, or whether…….” This sentence is too long and not very clear. Please rephrase it for better comprehension.

Author response: The following sentence has been rephrased from “There is debate as to whether discovery scientists should focus medicinal chemistry effort on synthesizing the most potent, efficacious and safe macromolecules and then rely on the delivery system to address sub-optimal PK characteristics after the fact, or whether to focus on optimization of the lead at the outset in respect of suitable physicochemical and stability features to cross epithelia intact” to “There is debate as to whether medicinal chemists in the discovery field should solely focus on safety and efficacy of the active, and rely on formulation and delivery scientists to address sub-optimal solubility, ADME and stability characteristics – or whether medicinal chemists to focus on all of the aforementioned properties to rely less on delivery and formulation scientists

Question 4: Line 87: What do you mean by “reductionist models”, In vitro cell culture models or others ? It would help to specify “reductionist models, such as………..”

Author response: Amended

Question 5: Line 93: is there a reference?

Author response: Reference included

Question 6: Line 205. “certainly” ??

Author response: Amended

Question 7: Line 215: “More recently, a structure activity assessment  of PIP-640 showed that residues….” 

Author response: Amended

Question 8: Line 419-20: “the permeation enhancement of anionic surfactants”

Author response: Amended

Question 9: Line 447. …was perfused rather…. than "perfusion"

Author response: Amended

Question 10: Line 549: “principle” do you mean “primary” ?

Author response: Amended

Question 11: Lines 579-80:   “due to a lower of intestinal fluid for dilution.”  Better would be:.... due to a lower amount of rectal fluid than in other intestinal segments.

Author response: Amended

Question 12: In Table 1: substitute sCT with salmon Calcitonin; use twice daily or thrice daily in place of the abbreviations bd and tds.

Author response: Amended

Question 13: Line 214: FD4 First time should be spelt out as Fluorescein isothiocyanate-dextran and MW range

Author response: Amended

Question 14: Line 244 use abbreviation sCT

Author response: Amended

Question 15: Line 342: what does PLGA stand for?

Author response: Amended

Question 16: Line 462: here I think you mean “human intestinal fluid” not SIF

Author response: Amended

Question 17: Line 466: use abbreviation SIF

Author response: Amended

Question 18: line 349: However,

Author response: Amended

Question 19: line 370: the rationale being that, had the effect….., the absorption….

Author response: Amended

Question 20: Line 467: …mirror dissolution media), or whether...

Author response: Amended

Question 21: Line 583: Nonetheless, time…

Author response: Amended

Question 22: Line 612: ….noteworthy  however, that…..

Author response: Amended

Question 23: Line 662: Hence, translation….

Author response: Amended
